# Structural and Functional Deviations of the Hippocampus in Schizophrenia and Schizophrenia Animal Models

**DOI:** 10.3390/ijms23105482

**Published:** 2022-05-13

**Authors:** David Wegrzyn, Georg Juckel, Andreas Faissner

**Affiliations:** 1Department of Cell Morphology and Molecular Neurobiology, Faculty for Biology and Biotechnology, Ruhr-University Bochum, Universitaetsstr. 150, D-44801 Bochum, Germany; david.wegrzyn@rub.de; 2Department of Psychiatry, LWL University Hospital, Ruhr-University Bochum, Alexandrinenstr. 1, D-44791 Bochum, Germany; g.juckel@lwl.org

**Keywords:** schizophrenia, hippocampus, overexcitability, interneurons, GABA, parvalbumin, perineuronal nets, incomplete inversion patterns, glutamate, extracellular matrix

## Abstract

Schizophrenia is a grave neuropsychiatric disease which frequently onsets between the end of adolescence and the beginning of adulthood. It is characterized by a variety of neuropsychiatric abnormalities which are categorized into positive, negative and cognitive symptoms. Most therapeutical strategies address the positive symptoms by antagonizing D2-dopamine-receptors (DR). However, negative and cognitive symptoms persist and highly impair the life quality of patients due to their disabling effects. Interestingly, hippocampal deviations are a hallmark of schizophrenia and can be observed in early as well as advanced phases of the disease progression. These alterations are commonly accompanied by a rise in neuronal activity. Therefore, hippocampal formation plays an important role in the manifestation of schizophrenia. Furthermore, studies with animal models revealed a link between environmental risk factors and morphological as well as electrophysiological abnormalities in the hippocampus. Here, we review recent findings on structural and functional hippocampal abnormalities in schizophrenic patients and in schizophrenia animal models, and we give an overview on current experimental approaches that especially target the hippocampus. A better understanding of hippocampal aberrations in schizophrenia might clarify their impact on the manifestation and on the outcome of this severe disease.

## 1. Introduction

In 1893, Emil Kraepelin described schizophrenia as a progressive neurodegenerative disease which leads to an irreversible loss of cognitive functions, and he differentiated it from manic depression [1]. While Kraepelin initially used the term dementia praecox, Eugen Bleuer renamed it to the commonly used designation schizophrenia [2]. Nowadays, it is known that schizophrenia is a complex neuropsychiatric disease with a mean lifetime prevalence of 1%, a reduced life expectancy and a multifactorial cause [3,4,5]. The recent DSM-5 criteria for schizophrenia include delusions, hallucinations, disorganized speech, a grossly disorganized or catatonic behavior and negative symptoms; two or more of these must persist for a period of one month or longer [5]. While schizophrenia is frequently diagnosed in early adulthood, the current state of research posits that the pathogenesis begins in early neurodevelopment [5,6]. This is supported by studies that have observed childhood neurobehavioral deficits in the offspring of schizophrenic parents [7]. In general, the symptoms of schizophrenia can be classified into positive, negative and cognitive symptoms. While the positive symptoms of schizophrenia include hallucinations, delusions, thought disorder and a lack of insight [8], the negative symptoms consist of social withdrawal, poverty of speech, self-neglect, highly reduced motivation and anhedonia [9]. Furthermore, cognitive symptoms are characterized by deficits in attention, working memory, verbal declarative memory and several other executive function impairments [10]. The brain of schizophrenic patients shows structural deviations that include, inter alia, a thinning of the cerebellar cortex [11,12] and a reduced volume of the thalamus and the striatum, accompanied by an enlargement of the lateral ventricles [13,14,15,16]. With a focus on cognitive symptoms that are impairing the life quality of patients, structural aberrations of the hippocampus might especially be of great interest. The hippocampus is located in the allocortex and can be subdivided into the dentate gyrus and the cornu ammonis (CA). It is strongly involved in higher cognitive functions like spatial memory, episodic memory, short-term memory and long-term memory [17]. Interestingly, schizophrenic patients show morphological as well as functional deviations of the hippocampus, as the following review will address. Furthermore, current findings of schizophrenia animal models will give an overview on recent approaches that target the hippocampus as a plastic structure of the central nervous system (CNS).

## 2. Structural Abnormalities of the Hippocampus in Schizophrenia Patients

For a long while, it has been well-known that structural aberrations of the hippocampus are a common hallmark of schizophrenia (Figure 1). Over decades, the evidence of severe hippocampal abnormalities in schizophrenia patients is mounting and includes a remarkable reduction of the hippocampal volume [18,19,20,21,22,23,24,25], a dysregulation of synaptic protein levels [26,27,28] and a dysconnectivity of the hippocampus from further parts of the CNS [29,30,31,32]. The general reduction of the hippocampal volume can already be observed in patients with a high risk for psychosis [33]. This indicates that the structural deviation occurs in a period before the disease manifestation, supporting the neurodevelopmental model for schizophrenia [34,35,36,37].

While former studies mainly focused on total volume changes of the hippocampus in schizophrenic patients, modern high-resolution magnet resonance imaging (MRI) techniques were utilized for a more detailed and region-specific analysis. Interestingly, these techniques contributed to the observation of so-called incomplete inversion patterns which are characterized by a round and verticalized hippocampal morphology with a deep collateral fissure, and a medial positioning in the coronal plane [38,39,40]. Surprisingly, this structural deviation can be observed in 18–19% of healthy subjects, predominantly in the left hemisphere, and was primarily suspected to contribute to the pathology of epileptic seizures [40,41]. Nevertheless, a recently published study showed that incomplete inversion patterns occur, on the one hand in a more severe manner and on the other hand with a higher frequency in schizophrenic patients [38]. In addition, this study unraveled a link between incomplete inversion patterns and the commonly described reduction of the total hippocampal volume, as well as of the increased volume asymmetry between the left and right hemispheres. Furthermore, the question of whether or not specific incomplete inversion patterns correlate with auditory and visual hallucinations was addressed. Intriguingly, schizophrenic patients with both auditory and visual hallucinations showed a flatter hippocampal morphology in the left hemisphere than healthy subjects and patients suffering exclusively from auditory hallucinations [42].

These observations highly support the neurodevelopmental model for schizophrenia since the inversion of the dentate gyrus and the cornu ammonis occurs in the second trimester of pregnancy [43,44,45]. Ultrasound studies could show that in approximately 50% of preterm neonates, the inversion of the hippocampus is not completely accomplished up to gestation week 24 [45]. However, at gestation week 25, the frequency of incomplete hippocampal inversions is comparable to the frequency of incomplete inversions in the adult population [45]. A genome-wide association study analyzed the heritability of incomplete inversion patterns in adolescents [46]. Here, a high heritability rate was observed with a significantly linked locus identified on chromosome 18q11.2. In addition to this genetic aspect, the second trimester of pregnancy, when the inversion of the hippocampus occurs, appears to be sensitive for environmental factors like inflammation, stress and anxiety [47,48,49]. For this period, it has been described that maternal stress and anxiety clearly have negative effects on neurodevelopmental events and induce structural changes of the hippocampus, as well as a general reduction of the grey matter volume [48,49]. While structural alterations of the hippocampus have been described in very preterm born infants, there is a lack of scientific data about the impact of pre- or perinatal infections and other complications with regard to the disruption of the hippocampal inversion process [50]. Since it is known that a very preterm birth and a low birth weight are risk factors for schizophrenia in later life, it would be of great interest to answer the question of if and how environmental risk factors affect the restriction of the hippocampal inversion process [51,52,53,54]. The identification of potential risk factors could contribute to the prevention of this morphological abnormality. The genetic and environmental risk factors which are suspected to be responsible for the deviations of the hippocampal morphology will be discussed in more detail at a later point of this review.

Despite the restriction of the hippocampal inversion and the general volume reduction, region-specific aberrations could be seen in patients with a high risk for psychosis and in schizophrenia patients. These were especially shown in the anterior and midbody cornu ammonis region 1 (CA1) and 2 (CA2). Narr and colleagues observed that mid- to antero-lateral hippocampal regions show a distinct neuroanatomical volume change in schizophrenic patients and proposed that these regions are more susceptible for the disease progression [55]. In contrast, another study showed a heterogeneous loss of the hippocampal volume at the posterior ends in early-onset schizophrenia (EOS) patients, while the main body volume of the hippocampus was increased [56]. Similarly, childhood-onset schizophrenia patients developed a morphological bilateral inward deformation of the anterior hippocampus which was furthermore related to the severity of the positive symptoms [57]. In this study, healthy siblings of schizophrenic patients were included and tended also to an anterior inward deformation, indicating a genetic vulnerability for this region-specific structural abnormality. The differing region-specific abnormalities of the hippocampus described by various studies might be explained by individual patient-specific risk factors that contributed to the disease manifestation (e.g., genetic factors, prenatal infections, perinatal complications, childhood abuse, drug abuse) or by the individual stage of disease progression (prodromal, acute, residual, childhood-onset). Based on these observations, the morphology of the hippocampus was suspected to give an overview on the disease progression and to be considered for the determination of individual therapeutical strategies. The interesting question is whether or not the volume reduction of the hippocampus in high-risk patients can be consulted for the prediction of a transition towards psychosis and was addressed in recent meta-analyses [58,59]. Importantly, both studies showed no statistical significance between the effect of the hippocampal volume on the transition risk for psychosis. Consequently, the volume of the hippocampus alone is not a sufficient predictor for the transition toward psychosis. Nevertheless, further functional parameters might be considered in addition to the morphological aberrations, as described in the following section of this review.

In addition to the macrostructural alterations of the hippocampus in schizophrenia patients, there are further differences on a synaptic level [60,61]. Here, a recently published study utilized a positron emission tomography (PET) radioligand for the synaptic vesicle protein 2A (SV2A) and observed a significant reduction of synaptic vesicles in the hippocampus of schizophrenia patients [60,62]. In addition, the postsynaptic compartment is affected in a structural manner. A significantly reduced density of dendritic spines in the prefrontal cortex was described in several *postmortem* studies and additionally confirmed in a recently published meta-analysis [63,64,65,66,67]. While the dendritic spine pathology was frequently described in the prefrontal cortex, less studies focused on dendritic spines in the hippocampus. Interestingly, the above-mentioned meta-analysis observed that the overall size of the effect of studies on the density of postsynaptic elements was unchanged in the hippocampus of schizophrenic patients [63]. Nevertheless, a reduction of the postsynaptic density protein-95 (PSD-95) could be verified on a protein level in the CA1-region and in the dentate molecular layer in *postmortem* studies [26,28]. Similar results were obtained for the synapse associated protein of 102 kDa (SAP-102) [68]. Contrary to these observations, a quantitative analysis of Golgi-stained hippocampal neurons showed an increase in the spine density on CA3 pyramidal cell dendrites and a rise in the number of thorny excrescences [27]. These observations indicate a necessity of further studies that characterize the dendritic spine densities and morphologies in the hippocampal subfields of schizophrenia patients. Table 1 contains an overview on the different types of structural aberrations in the hippocampus of schizophrenia patients with the corresponding references. 

## 3. Functional Aberrations of the Hippocampus in Schizophrenia Patients

The distinctive feature of total and region-specific volume changes, as well as of a functional dysconnectivity in schizophrenia patients, is frequently accompanied by an electrophysiological and metabolic hyperactivity of the hippocampus. Here, raised levels of glutamate and glutamine were already detected in patients with a high risk for psychosis via functional MRI analysis, as a previously published study can show [69]. Functional MRI studies of schizophrenia patients revealed a hippocampal hyperactivity during resting stages and during minimal cognitive tasks [70,71]. Furthermore, there is the evidence that the hippocampal hyperactivity spreads from a hypermetabolic CA1-region to the subiculum, inducing the transition from a prodromal state toward a psychotic state in patients with psychosis [72]. The hyperactivity of the anterior CA1-region and the subiculum correlates with the severity of positive, negative and cognitive symptoms [70,72,73]. With regard to cognitive parameters, the hippocampal activity was analyzed during tasks addressing specific parameters. Here, a higher hippocampal activation could be observed when schizophrenia patients performed sensory gating tasks or when they were merely exposed to urban noise stimuli [74,75]. Furthermore, schizophrenic patients developed hippocampal deficits in relational memory formation tasks [76,77], as well as face habituation tasks [78,79]. A hyperactivity of the hippocampus frequently goes along with a higher metabolic activity and cerebellar blood volume levels [80]. Measurements of the regional cerebellar blood flow levels in patients who are ultra-high-risk for psychosis showed a significant increase in comparison to healthy subjects [81]. The rise of the cerebellar blood volume levels interestingly vanished when the patients did not meet ultra-high-risk criteria [81]. Regional cerebellar blood volume analyses unraveled that the severity of delusions showed the highest association with increased blood volume levels in the CA1-region of the hippocampus [73]. The increase of the cerebellar blood volume levels in the CA1-subfield furthermore predicted the clinical progression from a prodromal state toward psychosis. While the volume reduction of the hippocampus alone is not sufficient to predict a transition from the prodromal state toward the psychotic state, cerebellar blood volume parameters could be additionally considered to predict an immediate psychotic phase. 

In addition to the hyperactivity of the hippocampus in patients at high risk for psychosis and schizophrenia patients, a functional dysconnectivity has been described for the hippocampo-striatal and the hippocampo-prefrontal pathway [29,31,82]. In a recently published study, the connectivity of the anterior hippocampus in patients with a first-episode psychosis is therefore discussed as a useful predictor for the response of an antipsychotic treatment [83]. Another study observed a functional hypoconnectivity of the hippocampus with regions that are involved in episodic memory, such as the medial prefrontal cortex or the parahippocampal gyrus [84]. In addition, functional MRI-scans of unaffected first-degree relatives of schizophrenia patients revealed a hyperconnectivity of the left anterior cingulate cortex with the right hippocampus and a hypoconnectivity of the right anterior cingulate cortex with the right hippocampus, in comparison to healthy controls [85]. This additionally indicates a familiar risk for the schizophrenia susceptibility. The investigation of unaffected first-degree relatives of schizophrenic patients is of great interest since it might draw attention to genetic or environmental factors which increase the susceptibility to this disease. As previously mentioned, individual hippocampal activity levels and connectivity measurements might be useful for therapy options and treatment responses, in addition to the volumetric data and cerebellar blood volume levels. The following figure gives an overview on the most common functional alterations in the hippocampus of schizophrenia patients (Figure 2).

## 4. Genetic and Environmental Risk Factors Induce Hippocampal Deviations

Although morphological abnormalities of the hippocampus in schizophrenia patients and schizophrenia animal models are well-known, it is still unclear as to how these manifest. A genetic predisposition seems to be a plausible explanation since studies proved a higher risk for the disease manifestation in twin studies [86,87]. Furthermore, the above-mentioned incomplete hippocampal inversion patterns of schizophrenic patients show a high level of heritability [46]. 

Modern transcriptome techniques were used to investigate the hippocampus of *postmortem* schizophrenia patients with regard to the up- and downregulation of specific genes. Interestingly, an extensive hippocampal transcriptome-analysis was performed in human schizophrenia patients and revealed subfield-specific dysregulations of schizophrenia-associated genes [88]. Here, an enrichment of excitatory neuronal and synaptic genes (e.g., *LMBRD2*, *MAL2*, *DLG3*, *MAP2*, *VDAC1*) was detected in the so-called CA3-M2 module via weighted gene co-expression analysis (WGCNA). The results indicate that the hippocampal CA3-region is genetically affected and possibly contributes to the hippocampal hyperactivity which is frequently described in schizophrenic patients. In addition, another identified module of this study, the so-called CA3-M3 module, included genes associated with microglia (e.g., *CSF1R*, *DOCK2*, *SYK*, *APBB1IP*), autism (*CUL3*, *GRIA1*, *BICD1*), and further neuron-specific genes (e.g., *SLC8A1, CNRIP1*, *KCNMB2*). Importantly, this study also considered the impact of an antipsychotic treatment on the gene expression levels in the hippocampus and observed that 80 genes in the dentate gyrus, 351 genes in the CA3-region and 188 genes in the CA1-region were differentially expressed between treated and untreated patients [88]. Based on this observation, future studies should take into consideration that treatment with antipsychotics clearly affects gene expression and might falsify the conclusions which are made based on the data. A further study precisely focused on the transcriptomic profiling of the granule cell layer in the human dentate gyrus of *postmortem* schizophrenia patients by laser-capture microdissection, followed by RNA-sequencing [89]. This study identified a different set of schizophrenia-associated genes which were not found in previous studies that utilized the complete hippocampus, including the calcium voltage-gated channel subunit alpha1C (*CACNA1C*) and the glutamate metabotropic receptor 3 (*GRM3*). Further transcriptome studies of past years have proven that altered gene expression levels in different brain regions of schizophrenia patients underline the genetic aspect of this disease [89,90,91,92,93,94]. The most recent of these analyses could interestingly show an altered expression of the γ-aminobutyric acid-A (GABA-A) receptor-subunits *GABRA1*, *GABRA2* and *GABRB3* in the dorsolateral prefrontal cortex and in the superior temporal gyrus [91,93]. These observations may increase interest on the role of GABA-A receptor-subunits and their role in schizophrenia. Further evidence of a genetic predisposition for hippocampal alterations is provided by the investigation of unaffected first-degree relatives of schizophrenia patients with distinctive hippocampal features, as previously mentioned [57,85]. 

Despite the genetic risk for the development of schizophrenia, there is growing evidence that environmental factors and inflammatory processes contribute to the manifestation of psychosis and hippocampal aberrations.

A prominent and well-known environmental risk factor is the experience of childhood trauma, which is highly linked to structural changes of the hippocampus in schizophrenia patients [95,96]. In a recently published study, a significant correlation between childhood trauma scores and volume changes across hippocampal subregions could be observed in a sex-dependent manner with significant effects experienced in women [97]. A similar connection between childhood trauma and a reduced hippocampal volume was observed in cases of major depression and posttraumatic stress disorder, with predominant deviations in the left hippocampus [98,99,100]. Furthermore, experiencing childhood maltreatment can induce a volume reduction of the hippocampal CA3-region, of the dentate gyrus and of the subiculum [101]. Here, an inhibition of the hippocampal neurogenesis caused by chronic stress and the release of adrenal steroids induced by the childhood trauma or maltreatment is discussed as a possible mechanism for the hippocampal abnormalities [102].

Interestingly, a current study revealed that the volume of the hippocampus and the amygdala—which are stress-sensitive structures—is affected by the grade of severity of adverse childhood experiences, especially during the periods of pre-adolescence and early adolescence [103]. In hippocampal lesion experiments with rhesus monkeys, it has been shown that early trauma induces effects similar to the symptoms of schizophrenic patients and an altered functional hippocampo-prefrontal network integrity [104,105]. Additional environmental factors that show an impact on the hippocampal morphology are maternal stress and maternal anxiety during mid-gestation, as initially mentioned in the first part of this review [48,49]. A currently published study analyzed the influence of maternal stress at different trimesters of the pregnancy and performed connectivity measurements of the hippocampus with further CNS structures [106]. Interestingly, this study could demonstrate that maternal distress in the third trimester of pregnancy was linked to a weaker hippocampal–cingulate cortex connectivity and to a stronger hippocampal–temporal lobe connectivity. Additionally, the same study revealed that increased cortisol levels in the second but not in the third trimester of pregnancy were associated with a weaker hippocampal–cingulate cortex connectivity and a stronger hippocampal–temporal lobe connectivity [106]. Therefore, the functional connectivity of the hippocampus can be negatively affected by stress, with varying effects depending on the point in time of the pregnancy when stressful periods occur. In studies with rhesus monkeys, it could be furthermore shown that prenatal stress significantly reduced the neurogenesis in the dentate gyrus of juvenile animals [107].

Based on this, genetic, as well as environmental factors seem to play an important role in the occurrence of hippocampal abnormalities in schizophrenia. This aspect might furthermore explain why some patients develop this neuropsychiatric disease without a familiar risk, and vice versa, why subjects do not develop psychosis even though they experienced trauma or similar negative experiences. Furthermore, cumulative effects of genetic and environmental factors may increase the severity of schizophrenia symptoms and impair the outcome of the disease.

## 5. Inflammatory Processes Contribute to Hippocampal Abnormalities

In past years, evidence has been raised that inflammation is another important key factor in the progression of schizophrenia. The presence of increased inflammatory pathways was proven through the detection of higher peripheral cytokine levels in schizophrenia patients [108]. Interestingly, inflammatory processes could be furthermore identified in the hippocampus of schizophrenic patients. Here, an extensive *postmortem* transcriptional profiling study revealed a robust enrichment of pro-inflammatory pathways in several parts of the CNS, including the prefrontal cortex, the hippocampus and the striatum [109]. The transcriptional profiling revealed an upregulation of signaling pathways associated with interleukin-6 (IL-6), signal transducer and activator of transcription 3 (Stat3), oncostatin and interferon, as well as a downregulation of signaling pathways associated with GABA-receptors, synaptic long-term potentiation and Ephrin-B [109]. Another study could detect an increase of inflammatory-related genes like interferon-induced transmembrane protein 1, 2 and 3 (*IFITM1,2,3*), apolipoprotein L1 (*APOL1*), adenosine receptor A2a (*ADORA2A*), insulin-like growth factor-binding protein 4 (*IGFBP4*) and a cluster of differentiation 163 (*CD163*) in the hippocampus of patients via mRNA-sequencing [110]. 

Since microglia are the resident immune cells of the CNS, the hypothesis arose that this cell type might contribute to the disease’s manifestation and progression by the release of inflammatory factors and an extensive synapse elimination [111]. Microglia were first described by Pio del Rio-Hortega in 1919 as the fourth element of the CNS [112]. During early embryonic development, microglia originate from erythromyeloid progenitors (EMPs) located in the extra-embryonic yolk sac and migrate into the CNS [113,114,115]. In the CNS, microglia form a stable population that undergoes slow proliferation and apoptosis. Nowadays, it is known that microglia contribute to the elimination of excessive synapses, to the dendritic maturation and to the postmitotic differentiation of neurons [116,117,118,119,120]. With regard to the role of the hippocampus for schizophrenia, microglia might be of greater interest because they show a regional heterogeneity with a higher density in the hippocampus compared to other regions of the CNS [121]. Furthermore, both neurons and microglia possess functional receptors for the released factors of the others [122,123,124,125,126,127,128]. Therefore, a bidirectional interaction between neurons and microglia exists as discussed by Szepesi and colleagues [129]. Recently published findings support this idea by demonstrating that microglia act as negative feedback regulators of neuronal activity and consequently protect the CNS from excessive activation [130]. A dysregulation of microglia might consequently increase the neuronal activity and serve as another possible explanation for the hyperactive state of the hippocampus in schizophrenia patients. On the other hand, a hyperactive neuronal network might vice versa affect the activation state of microglia and induce the release of inflammatory factors or matrix-degrading enzymes. 

Indeed, several studies observed an increased number or a higher activation state of microglia in the hippocampus of schizophrenia patients [131,132]. Interestingly, the analysis of schizophrenia patient-derived microglia-like cells revealed a higher synapse engulfment and elimination on neuronal cells in vitro [133]. In this context, important studies identified the major histocompatibility complex (MHC), and especially the complement system, as an important mediator of synapse elimination in schizophrenia [134,135,136,137]. While a higher reactive state of microglia could be a plausible explanation for the hippocampal deviations, positron emission tomography (PET) studies of recent years that targeted the translocator 18 kDa protein (TSPO) did not find significant—or found only minor microglial—changes in recent-onset or established schizophrenia patients [138,139,140,141]. However, there is increasing doubt that TSPO is a reliable marker for activated microglia since it is not correlated with other microglial activation markers and furthermore not restricted to microglia [142]. In addition, further PET studies with variable markers could verify activated microglia in the CNS of schizophrenic patients [143,144]. Considering these conflicting observations, a carefully validated marker for activated microglia is highly desirable for future PET studies and investigations. 

Notwithstanding, inflammation seems to be an important contributor to this neuropsychiatric disease. Therefore, several clinical studies considered this aspect and administered an additive treatment of patients with the neuroprotective and anti-inflammatory antibiotic minocycline. Surprisingly, opposing observations were made when an adjunctive treatment with minocycline was performed. While several studies proved beneficial effects for the negative and cognitive deficits, as well as for the pro-inflammatory cytokine-levels [145,146,147,148,149,150,151], other studies did not observe a benefit for schizophrenia patients [152,153]. These opposing results do not necessarily mean that minocycline has no beneficial effects. The point in time when minocycline is administered could be a critical aspect, as previously discussed by Kishimoto and colleagues [154]. Furthermore, the additive treatment with minocycline might be useful for a subgroup of schizophrenia patients that clearly show a rise of inflammatory pathways. Therefore, it could be useful to initially screen for inflammatory factors in blood samples before a minocycline treatment is planned. A summary on the previously mentioned risk factors is given in the following schematic depiction (Figure 3).

In addition to microglia, astrocytes are another glial cell type that might be of interest for the pathophysiology of schizophrenia. Here, various studies described region-specific differences either in the number of astrocytes or in the expression levels of astrocyte-specific markers like the glial fibrillary acidic protein (GFAP), the aldehyde dehydrogenase 1 L1 (ALDH1L1) or the glutamate transporter-1 (GLT-1) in *postmortem* tissue samples of schizophrenia patients [155,156,157]. Interestingly, a genome-wide association study identified six astrocyte gene sets that are strongly associated with schizophrenia, indicating a possible role of astrocytes for the disease’s manifestation and progression [158]. However, no significant differences were described concerning the properties of astrocytes in the hippocampus of schizophrenic patients. Here, the analysis of the astrocyte density in posterior hippocampal subregions of *postmortem* schizophrenia patients revealed no deviations [159]. In another *postmortem* study, a similar number of phosphorylated-GFAP (pGFAP) positive cells was observed in the hippocampus of healthy subjects and schizophrenic patients [160]. These observations are furthermore supported by a transcriptomic analysis that revealed no significantly altered expression profiles of astrocytes in subcortical regions, including the hippocampus [161]. Although astrocytes seem to be unaffected in the hippocampus of *postmortem* schizophrenia patients, their possible influence should not be underestimated since a strong impact of genetically modified astrocytes on hippocampal neurons could be shown. Here, the use of an astrocyte-specific mutant version of the disrupted in schizophrenia 1 gene (*DISC1*) significantly reduced the proliferation rate of neural progenitors and the dendrite growth of newborn neurons in the adult hippocampus [162]. These observations were furthermore associated with elevated levels of anxiety, attenuated social behaviors, and impaired hippocampus-dependent learning and memory [162]. Therefore, additional astrocyte-specific studies are necessary to fully clarify the role of this glial cell type for schizophrenia.

## 6. Structural and Functional Hippocampus Deviations in Animal Models for Schizophrenia

Different animal models were established to shed light on the mechanisms responsible for the pathophysiology of schizophrenia. As previously mentioned, prominent region-specific effects were observed in the human anterior hippocampus that corresponds to the ventral hippocampus in rodents. The structural organization of the hippocampus in mice and rats is very similar, as a recent flatmap study has shown [163]. Here, a spatial similarity was demonstrated by rescaled comparisons. Furthermore, a similar distribution of parvalbumin-positive interneurons in the murine and rat hippocampus was shown [164]. Nevertheless, it has been observed that there are more adult-born hippocampal neurons in rats than in mice that interestingly mature faster and are more strongly associated with animal behavior [165]. On a functional level, the hippocampal formations of mice and rats show similar macroscopic physiological patterns with a similar spatial and regional distribution of extracellular voltage changes, frequencies of oscillations and correlations between local field activities and animal behaviors [166]. However, significant differences were observed with regard to the place cell activities in the murine and rat hippocampus [167]. These observations should be considered when newly generated data are interpreted and discussed in future studies. Although the majority of schizophrenia animal models are based on rodents, there is a growing interest in studies also utilizing the zebrafish as a model organism [168]. Since the brains of zebrafish do not contain a typical hippocampal formation, this organism will not be discussed in more detail within the scope of this review. However, a recently published study performed a phenotypical landscaping of schizophrenia-associated genes in the zebrafish and revealed an association of some risk genes with a lowered volume of the pallium, the homologue structure of the hippocampus in mammals [169,170]. The following section reviews the findings of established schizophrenia models, including (1) prenatal immune activation or prenatal neurotoxicity animal models, (2) studies with treatment-induced symptoms related to schizophrenia and (3) knockout models and current optogenetic approaches.

Maternal immune activation (MIA) models for schizophrenia are based on epidemiological studies that unraveled a link between prenatal infections during influenza pandemics and a higher occurrence of schizophrenia cases in the adult offspring [171,172,173,174,175,176]. Pregnant mice treated with a sublethal dose of the human influenza virus gave birth to animals with a significant upregulation of the schizophrenia and autism related gene Forkhead box P2 (*Foxp2*) in the hippocampus at P35 and P56 [177]. This was accompanied by an atrophy of different brain regions and an altered gene expression in the hippocampus of offspring animals [177]. Interestingly, additional studies showed that further pathogens can also increase the risk for schizophrenia in the offspring [178,179,180,181,182,183]. Here, infections during the second trimester of pregnancy were especially proven to increase schizophrenia risk. The varying trimester-dependent effects of the maternal immune challenge on CNS development and on behavioral abnormalities in the offspring were furthermore observed in animal models [184,185,186]. Since the placenta is impenetrable for most pathogens, the induction of the maternal immune response and the associated release of cytokines is highly suspected to be responsible for the previously mentioned effects [187,188,189,190]. Therefore, MIA models for schizophrenia frequently induce an activation of the maternal immune response by prenatal injections with immunostimulants like polyinosinic–polycytidylic acid (Poly I:C) or lipopolysaccharide (LPS). The most common cognitive and behavioral abnormalities in these models were observed for parameters like prepulse inhibition [191,192,193,194,195,196,197], latent inhibition [195,198] and social interaction [193,199,200]. In rhesus monkeys, a Poly I:C-induced maternal immune challenge increased repetitive behavior patterns and reduced communication, as well as social interactions in the offspring [201]. Furthermore, a smaller average soma size of neurons, which was, however, not significant, could be observed in a prenatal maternal immune activation study on rhesus monkeys [202].

Interestingly, the specific analysis of the hippocampus in prenatal immune activation models revealed structural and functional abnormalities which share similarities with the hippocampal deviations in schizophrenic patients. The activation of the maternal immune system via the administration of Poly I:C at gestation day 15.5 in pregnant mice induced a significantly reduced hippocampal volume, a decreased number of parvalbumin-positive interneurons and a lowered synaptic inhibition in mature dentate gyrus neurons of three-month-old male offspring animals [203]. Furthermore, the induction of a maternal immune challenge with Poly I:C decreased the serotonin and taurine levels in the hippocampus of offspring animals and affected neurotransmitter levels in other parts of the CNS [204]. On a functional level, the treatment of pregnant rat dams with LPS at embryonic day 15 and 16 led to smaller evoked field excitatory postsynaptic potentials (fEPSPs), but to a higher intrinsic excitability of CA1 pyramidal neurons in hippocampal slices of 20–25 day-old offspring animals [205]. Similarly, a reduced firing frequency and increased amplitudes of miniature excitatory postsynaptic currents (mEPSPs) in the hippocampal CA1-region of mice derived from Poly I:C- treated dams could be observed in hippocampal slices [206]. In addition, dissociated hippocampal neurons isolated from embryonic mice of Poly I:C-treated dams developed a network hyperactivity and disruptions of perineuronal nets (PNNs), as well as reduced soma areas of PNN-wearing neurons in vitro [207]. PNNs are a specialized and condensed form of the extracellular matrix (ECM) that stabilize synaptic contacts but prevent the formation of new connections [208]. Finally, a study that induced an immune challenge in the early postnatal phase instead of the prenatal phase could observe hippocampal long-term effects and an impairment of learning and memory [209]. 

Based on these observations, different studies addressed the question of whether or not the effects which were induced by a maternal immune activation can be prevented or ameliorated. Here, beneficial results were achieved when offspring animals received a treatment with minocycline [210,211], different neuroleptics [212] or IL-6-blocking antibodies [195]. Anti-inflammatory agents might be of great interest since a microglial activation could be shown in a Poly I:C mouse model for schizophrenia [213]. Regarding the role of IL-6 for the structural integrity of the hippocampus, a recently published study could demonstrate that there is a correlation between increased levels of IL-6 and a decrease of the total hippocampal volume [214]. Additionally, it has been shown that IL-6 can pass the placenta more efficiently during the mid-gestation period than during the late-gestation period and induce the release of fetal stress hormones [215]. A currently published study analyzed if the effects of a prenatal maternal immune activation can be enhanced by an environmental enriched housing during the experimental procedures [216]. This study provided evidence that the expression of the stress-associated markers oxytocin receptor (*Oxtr*), corticotropin-releasing hormone (*Crh*) and nuclear receptor subfamily 3 group C member 1 (*Nr3c1*) was significantly increased in the ventral hippocampus of MIA offspring animals. Regarding the oxytocin receptors in the hippocampus, it has been shown that neonatal stress negatively affects the development of this system in rats [217]. Furthermore, increased levels of corticosterone were identified in plasma specimens in offspring animals, indicating a dysregulation of the hypothalamic–pituitary–adrenal (HPA)-axis [216]. The housing in an environmentally enriched surrounding reversed the MIA-induced effects in the offspring [216]. In conclusion, different independently performed studies observed structural, as well as functional deviations of the hippocampus in the offspring of dams exposed to maternal immune activation. This aspect and new preventive therapies might be of great interest with regard to the COVID-19 pandemic and its possible consequences regarding neuropsychiatric disorders in the future. 

In contrast to maternal immune activation models, the prenatal methylazoxymethanol acetate (MAM)-model directly induces neurotoxic effects and is consequently a developmental disruption model for schizophrenia [218]. Interestingly, studies utilizing the MAM-model revealed a reduction of the hippocampal volume in the offspring with schizophrenia-related behavioral abnormalities [219,220]. Proteomic, as well as metabolomic analyses produced strong evidence that a treatment of rats with MAM at embryonic day 17 is responsible for deficits primarily in the hippocampus, affecting the glutamatergic neurotransmission [221]. Here, electrophysiological recordings of the CA1-region in the hippocampus revealed a significant reduction of fEPSPs and a deficit of the AMPA-receptor-mediated synaptic transmission. Furthermore, animals that received a prenatal MAM-treatment developed hyperactivity in the ventral hippocampus in later stages of development [222,223]. On a cellular level, a previously published study could identify a loss of parvalbumin-positive interneurons in combination with a reduced gamma-band response [224]. Interestingly, an additional published study showed that a MAM-treatment at gestational day 16 and 17 induced a reduction of the hippocampal volume, an impaired contextual fear memory and a significantly reduced long-term potentiation (LTP) in the synapses of the CA1-region in mice [225]. Furthermore, this study focused on sex-specific differences and could demonstrate that male offspring mice that previously suffered from a MAM-treatment at gestation day 16 developed a decreased parvalbumin-expression in the hippocampus and deficits in the delayed alternation task. The loss of parvalbumin-positive interneurons in the hippocampus of MAM-treated animals was also observed in an additional study and could, interestingly, be prevented by a peripubertal treatment with diazepam [226]. The peripubertal treatment with diazepam prevented the hyperresponsivity of the dopamine system in the MAM-model [227]. As previously described for prenatal immune activation models, the housing of animals in an environmentally enriched surrounding within a prepubertal time window prevented dopamine dysregulation and hippocampal hyperactivity in a murine MAM-model [228]. The experiments that utilized an enhanced environmental enrichment might be interesting for new preventive therapeutical approaches. 

In the ketamine model for schizophrenia, acute psychosis is induced by injections in rodents. Here, ketamine acts as a N-methyl-D-aspartate receptor (NMDAR) antagonist and induces behavioral abnormalities which are similar to the positive and negative symptoms of schizophrenia patients [229,230,231]. Similar deviations were observed when other NMDAR antagonists, like phencyclidine or MK-801 were given, as well as in cases of autoimmune anti-NMDA receptor encephalitis [232,233,234,235]. A treatment of animals with ketamine showed strong effects on the hippocampal function and structure. Here, a repeated exposure to ketamine shifted the hippocampus towards a hypermetabolic basal state with an accompanying atrophy and a disruption of parvalbumin-expressing interneurons [72]. Interestingly, a currently published study analyzed hippocampal–prefrontal cortex (HPC-PFC) local-field potentials before and after the administration of ketamine [236]. The results showed that ketamine promotes an abnormal delta-high-gamma cross-frequency-coupling in the PFC and a rise of responses in the hippocampus. Furthermore, the authors could observe that LTP-induction before the ketamine treatment prevented the increase of the gamma amplitude [236]. Therefore, a rise of the glutamatergic synaptic efficiency is suspected to enhance cognitive impairments in animal models for psychosis, as discussed by the authors. Furthermore, it has been described that a subclass-imbalance of parvalbumin-expressing GABAergic interneurons occurs in the ketamine mouse model with disturbances of PNNs [237]. The disruption of PNNs in the ketamine model for schizophrenia was supported by a high-resolution confocal microscopic study that observed a significantly altered PNN fine-structure in a ketamine rat model [238]. In this study, ketamine-treated animals developed a significantly reduced mean area of PNN-units and so-called ECM-enriched vertices [238]. Although this study focused on the deep layers IV and V of the prelimbic cortex, similar PNN alterations might also occur in the hippocampus since it is a highly ketamine-affected structure. The role of parvalbumin-positive PNN-carrying neurons will be considered in more detail in a later paragraph of this review. Besides the manipulation of the glutamatergic synapse transmission, both electrophysiological or other pharmacological NMDAR-stimulation of the hippocampus induced a disruption of the prepulse inhibition in rats [239,240]. As already indicated previously, an impairment of prepulse inhibition can be frequently seen in schizophrenic patients, which is caused by sensory gating deficits [241]. Here, the activity level of the ventral hippocampus seems to play an important role in this deviation [242]. 

Studies of recent years that utilized knockout models and optogenetic manipulations allowed for the analysis of subfield-specific, as well as receptor-specific mechanisms which possibly contribute to the schizophrenia-related hippocampal alterations.

Interestingly, a subfield-specific *GluN1*-knockout mouse model with a disease-like disruption specifically in the dentate gyrus of the hippocampus showed several functional abnormalities [243]. Here, a psychosis-like behavior could be observed which was accompanied by a CA3-specific hyperactivity and an increased glutamate transmission at synapses between mossy fibers and CA3 neurons. Furthermore, an increased number of cFos-activated pyramidal neurons in the CA3-region could be seen [243]. Another recently published study revealed that an early ablation of corticolimbic NMDA receptors on interneurons induces a significantly reduced functional connectivity between the ventral hippocampus and the prefrontal cortex before and after adolescence [244]. These observations highly underline the important role of glutamate receptor subunits for psychosis, as well as for schizophrenia. Other studies investigated cyclin D2 mutant mice (*Ccnd2*) and observed deficits in parvalbumin-positive interneurons and a raised hippocampal excitability, including increased metabolic activity [245]. This study also described cognitive impairments when cyclin D2 was missing.

As previously indicated, optogenetics can be used for a region-specific manipulation of the neuronal activity and were discussed as a useful tool for the analysis of the disease-underlying mechanisms [246]. Interestingly, the optogenetic activation of excitatory neurons in the ventral hippocampus of mice induced a hyperlocomotion which is known as a rodent correlate of the positive symptoms of schizophrenia [247]. The optogenetic activation of the ventral hippocampus furthermore impaired performance on the spatial novelty preference test of short-term memory, indicating a cognitive impairment [247]. Conversely, the optogenetic inhibition of ventral hippocampal neurons enhanced motor learning dysfunction in a phencyclidine rat model for schizophrenia [248]. A currently published study utilized excitatory chemogenetic constructs in order to specifically induce a hyperactivation of mossy cells in the ventral dentate gyrus that are targeting granule cells and interneurons in the dorsal part of the dentate gyrus [249]. Here, the authors could show that a chemogenetic activation of the ventral mossy cells increased the activity of dorsal granule cells and significantly impaired the test performance in an object location memory task [249]. Considering these observations, the specific optogenetic manipulation of neuronal circuits emerges as a promising method which will unravel new aspects and important circuits of this neuropsychiatric disease in future studies.

## 7. Interneuron-Abnormalities in the Hippocampus of Schizophrenic Patients and Schizophrenia Animal Models

With regard to the electrophysiological hyperactivity, it has been proposed that parvalbumin-positive interneurons are hypofunctional, and this consequently results in an overstimulation of excitatory neurons [250]. Possibly, this idea might be adopted for the interneurons in the hippocampus. Interestingly, studies on *postmortem* schizophrenia patients unraveled a reduction of somatostatin- and parvalbumin-positive interneurons in the hippocampus, which was furthermore confirmed by qRT-PCR analysis [251,252,253]. As previously mentioned, a hippocampal transcriptome analysis also observed a significant reduction of GABAergic signaling pathways in schizophrenic patients [109]. Somatostatin- and parvalbumin-positive interneurons fulfill an important function by contributing to rhythmic gamma-oscillations in the CNS [254,255,256,257]. In schizophrenia patients, disturbances of these gamma-oscillations were frequently described by several independent studies [258,259,260,261,262]. Therefore, disruptions of interneurons might be the possible cause and inducer of the hippocampal hyperactivity in schizophrenic patients. The inhibition of the parvalbumin- and GAD65- positive interneuron population induced behavioral deficiencies like an impaired prepulse inhibition and startle reactivity [263]. With regard to therapeutical strategies, a recently published study investigated the gamma oscillations in schizophrenia patients and discussed them as a possible biomarker for the efficacy of targeted cognitive training (TCT) [264]. Here, the baseline of the gamma power predicted cognitive benefits after a full course of TCT. Furthermore, a change of the gamma power after 1 h of TCT predicted the enhancement of both positive and negative symptoms [264]. Interestingly, exercise in the form of treadmill running also had positive effects on the interneuron activation-dependent adult hippocampal neurogenesis in a MK801-induced schizophrenia-like animal model, as a current study demonstrated [265]. Additional interesting approaches of recent years focused on interneuron or interneuron precursor cell transplants in the hippocampus in different animal models for schizophrenia. Astonishingly, transplants consisting of stem-cell derived interneurons reduced the hippocampal hyperactivity and normalized the aberrant characteristics of dopaminergic neurons in a MAM-rat model for schizophrenia [266,267]. Similar results were achieved with interneuron-precursor transplants in adult *Ccnd2* knockout mice [268]. Here, Gilani and colleagues observed that mice developed a lowered metabolic hippocampal hyperactivity and an improved context-depending learning and memory up to six months after transplantation of the interneuron precursors [268]. Interestingly, the beneficial effects of interneuron transplants were also observed in an animal model for autism, as a currently published study could show [269]. Here, the social behavioral deficits were rescued by the transplantation of interneurons [269]. Furthermore, a prenatal immune activation model exhibited a reduction of parvalbumin-positive interneurons and an impaired GABAergic transmission in the dentate gyrus in offspring animals [270]. Here, a treatment with minocycline or with the microglia-specific arginase-1 prevented this effect. Interestingly, another current study revealed that a regular physical exercise significantly increased the neurogenesis of parvalbumin-positive interneurons with ameliorating effects for schizophrenia-like phenotypes [265]. These results show that targeting and manipulating interneurons in the hippocampus of schizophrenic patients could be a promising way to weaken the symptoms of this disease. Furthermore, this aspect might be especially interesting for cognitive impairments of schizophrenic patients which highly disturb life quality. A summary of beneficial experimental approaches with regard to hippocampal hyperactivity is given in Figure 4.

A subclass of parvalbumin-positive interneurons is covered by a dense and lattice-like structure of the ECM around the soma and proximal dendrites. These structures are the so-called perineuronal nets which were first described by Camillo Golgi more than a century ago [273]. Interestingly, perineuronal net-wearing neurons are represented in the hippocampus and can be observed, inter alia, in the CA1-, CA2- and CA3-subfield. Nowadays, several studies observed an important involvement of perineuronal nets for the regulation of the synaptic plasticity [271,274,275,276,277,278,279]. Furthermore, PNNs unfold a neuroprotective effect on parvalbumin-positive interneurons and fulfill ion-sorting and buffering functions [280,281,282,283]. Concomitant with the previously mentioned abnormalities of parvalbumin-positive interneurons, different studies showed disruptions of PNNs in schizophrenia patients. Here, *postmortem* analyses of schizophrenic patients especially revealed either a significant reduction in the amount of PNN-wearing neurons or in the PNN-staining intensity in the prefrontal cortex [284,285], in the amygdala [286,287] and in the inferior colliculus [288]. Contrarily, a currently published study did not find disturbances of parvalbumin-positive interneurons or PNNs in *postmortem* tissue of schizophrenia patients, although a negative correlation between the PNN density and the occurrence of psychosis could be shown [289]. Most of the human *postmortem* studies and animal models focused on PNN abnormalities in the cortex and unfortunately less is known about PNN alterations in the hippocampus of schizophrenic patients or schizophrenia animal models. Based on this, there are extensive reviews which focus on the role of perineuronal nets in schizophrenia [290,291,292,293,294]. Nevertheless, a currently published study utilized a double-hit model with a perinatal injection of the NMDAR antagonist MK801, combined with a post-weaning social isolation as an early-life stress event [295]. Interestingly, the authors could observe a region-specific reduction in the number of PNNs and PNN-wearing parvalbumin-positive neurons especially in the CA1-subfield of the hippocampus [295]. In addition, the presence of atypical perineuronal nets in the CA2-subfield was observed to be associated with a dysfunction of the social memory [296]. In this perspective, the targeting of perineuronal nets might be an interesting approach with regard to the functional deficits of the hippocampus in schizophrenia and schizophrenia animal models. The manipulation of perineuronal nets in the CNS was frequently performed by the stereotactic administration of bacterially-derived chondroitinase ABC (ChABC) or hyaluronidase in rodents [271,274,275,296,297,298,299,300]. Intriguingly, the experimental digestion of PNNs by ChABC-injections in the ventral hippocampus raised the general hippocampal activity, increased the activity of dopaminergic neurons and augmented the locomotor response to amphetamine indicating possible implications for schizophrenia [301]. With a focus on the hippocampus, as well as on cultured hippocampal neurons, electrophysiological alterations were observed when perineuronal nets were disrupted in vivo and in vitro [271,302,303,304,305,306]. Interestingly, there is a growing evidence that microglia are involved in the maintenance of perineuronal nets [307]. In disease models for Alzheimer’s and Huntington’s, a contribution of microglia to the loss of perineuronal nets could be proven [308,309]. Furthermore, an in vitro study revealed a disruption of perineuronal nets in cultured hippocampal neurons after an incubation with microglia conditioned medium which was accompanied by altered structural synapse numbers and electrophysiological network properties [305]. In contrast, a depletion of microglia induced more dense perineuronal nets [310]. As previously introduced, microglia and inflammatory processes are suspected to play an important role in schizophrenia. Therefore, the targeting of microglia might be an interesting option for reversing perineuronal net disturbances and restoring a synaptic balance.

While the procedure of PNN-digestion can be performed in animals, it is problematic for a therapeutical strategy in humans. However, a currently published study achieved a modulation of perineuronal nets by the oral administration of 4-methylumbelliferone (4-MU), an inhibitor of the hyaluronic acid synthesis [311]. Here, a reduction of perineuronal nets was accompanied by an enhancement of the memory retention when mice were treated for 6 months with 4-MU [311]. This approach might be interesting since it allows for the manipulation of perineuronal nets in the CNS by an oral treatment. However, in schizophrenia animal models, an increase and a strengthening of the PNNs is of greater interest than a disruption. Here, another mild approach could allow for new therapeutical strategies. Several studies could show that an enhanced environmental enrichment increased the density of perineuronal nets in the developing striatum [272], in the prefrontal cortex [312] and importantly in the CA2-subfield of the hippocampus [271]. As previously described in this review, current studies revealed that the housing in an environmentally enriched surrounding improved the deficits in different animal models for schizophrenia [216,228]. Therefore, the environmental enrichment could be used as a non-invasive method for the manipulation of perineuronal nets in regions of the CNS that are involved in the pathology of schizophrenia and allow for new therapeutic strategies. 

Based on this, interneurons especially play an important role for the functional deviations in the hippocampus of schizophrenic patients. Future studies might combine schizophrenia animal models with PNN-modulating agents or environmental enriched strategies to shed light on enhancing, as well as worsening effects for cognitive symptoms of schizophrenia or other neuropsychiatric, as well as neurological disorders.

## 8. Conclusions

In summary, hippocampal deviations are a common hallmark of the severe neuropsychiatric disorder schizophrenia. Although a majority of schizophrenia studies focus on the prefrontal cortex and the striatum in patients, hippocampal formation should not be underestimated since it contributes to higher cognitive functions. While former experiments have already unraveled a volume reduction of the hippocampus in schizophrenic patients, novel techniques allow for region-specific, morphological and functional analyses that provide a better overview on various hippocampal parameters in patients. Furthermore, hippocampal deviations, similar to those of schizophrenic patients, were observed in different animal models for schizophrenia. In recent years, several interesting experimental approaches were utilized in these models that targeted the hippocampus especially on a functional level. Here, a reduction of the hyperactivity went along with an amelioration of the behavioral and cognitive deficiencies in these models. In patients, hippocampal parameters could give valuable information about the disease’s progress, the risk of an immediate psychotic phase and the efficacy of different therapeutical strategies.

## Figures and Tables

**Figure 1 ijms-23-05482-f001:**
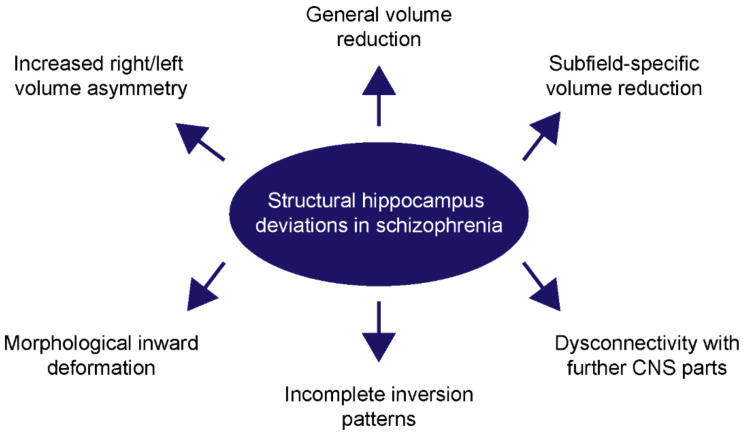
An overview of the most common structural hippocampus deviations in schizophrenia. Patients suffering from schizophrenia frequently develop a reduction of the general hippocampal volume, as well as a higher volume asymmetry between the left and right hippocampus. Furthermore, subfield-specific volume changes were especially described for the CA1- and CA2-region. Besides volume reductions, morphological abnormalities are known and include incomplete inversion patterns and inward deformations. Lastly, connectivity studies unraveled an impaired connection between the hippocampus of schizophrenic patients with further parts of the CNS.

**Figure 2 ijms-23-05482-f002:**
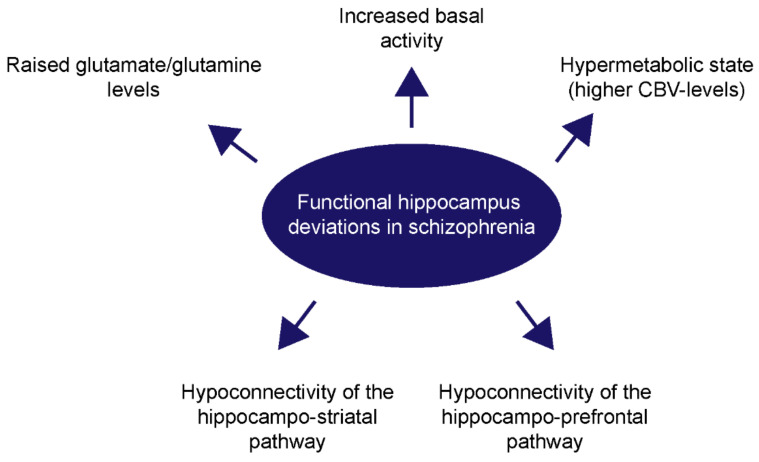
An overview of the functional deviations in the hippocampus of schizophrenia patients. In schizophrenia, the basal hippocampal activity is frequently increased and develops a hypermetabolic state with increased glutamate- and glutamine-levels and a higher cerebellar blood volume (CBV). Additionally, the hippocampo-striatal, as well as the hippocampo-prefrontal pathway, show a reduced functional connectivity.

**Figure 3 ijms-23-05482-f003:**
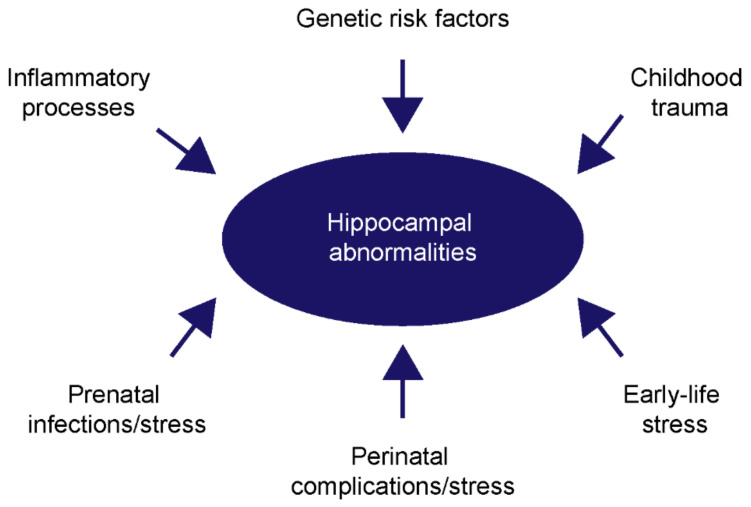
The possible risk factors for hippocampal deviations in schizophrenia. Besides genetic factors that increase the risk for hippocampal alterations, several environmental factors have been identified that directly influence hippocampal integrity. Here, especially prenatal, perinatal and early-life stress have negative effects regarding structural, as well as functional properties of the hippocampus. Furthermore, childhood trauma and inflammatory processes can negatively affect the development and the physiology.

**Figure 4 ijms-23-05482-f004:**
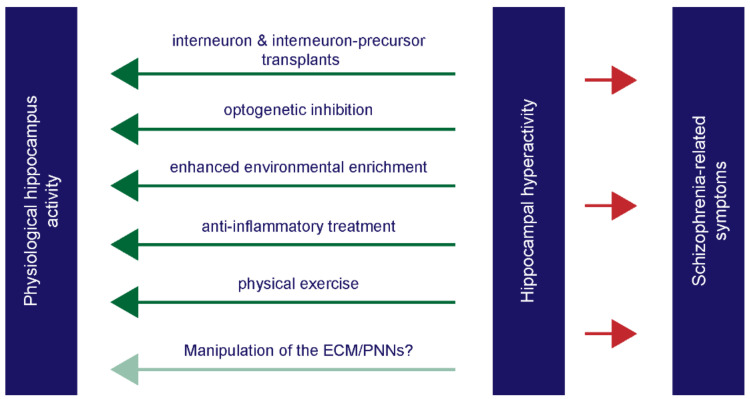
The intervening strategies that enhance hippocampal deviations in animal models for schizophrenia. Several studies targeted the hippocampus in schizophrenia animal models and observed ameliorating effects induced by different experimental approaches. Here, the transplantation of interneuron and interneuron-precursor transplants resulted in a reduction of the hippocampal hyperactivity [266,267,268]. Additionally, a reduction of the increased hippocampal activity could be achieved using optogenetic techniques [248]. Physical exercise [265], as well as an enhanced environmental enrichment [216,271,272] during the experimental procedures showed beneficial and protecting effects regarding the hippocampal activity. In maternal immune activation models for schizophrenia, a treatment with anti-inflammatory agents enhanced hippocampal parameters in the offspring [270]. Lastly, a manipulation of the extracellular matrix, especially of the interneuron-surrounding perineuronal nets, could be another possible approach for schizophrenia since this neuronal subtype shows disruptions in patients and goes along with impaired gamma-waves.

**Table 1 ijms-23-05482-t001:** An overview on the structural abnormalities of the hippocampus in schizophrenia patients.

Type of Structural Deviation	Experimental Approach	Reference
general volume reduction of the hippocampus	*postmortem* morphometric measurement	[18]
general volume reduction of the hippocampus	Coronal MRI/ high-resolution MRI	[19,21,22,55,56]
volume reduction of the left amygdala/hippocampal complex (AHC)	Coronal MRI/ high-resolution MRI	[23]
volume reduction of hippocampal subfields in the left hippocampus	Meta-analysis of *postmortem* studies	[25]
higher rate of incomplete hippocampal inversions with a reduced hippocampal volume	Coronal MRI/ high-resolution MRI	[38,39]
association of visual hallucinations with specific inversion patterns	Coronal MRI/ high-resolution MRI	[42]
volume reduction of anterior and midbody CA1- and CA2- regions with increased peri-hippocampal CSF levels	Coronal MRI/ high-resolution MRI	[20,55]
bilateral inward deformation of the anterior hippocampus	Coronal MRI/ high-resolution MRI	[57]
reduction of SV2A-positive synaptic vesicles in the hippocampus	PET-scan analysis	[60]
unchanged density of postsynaptic elements in the hippocampus	Meta-analysis of *postmortem* studies	[63]
reduced PSD-95 levels in the CA1-region/dentate molecular layer	Immunoblot-analysis of *postmortem* samples/ immuno-autoradiography	[26,28]
reduced SAP-102 levels in the hippocampus	Western blot analysis of *postmortem* samples	[68]
increased spine density and PSD-95 levels on CA3 pyramidal cell dendrites	Western blot analysis of *postmortem* samples/ Golgi staining	[27]
abnormal resting-state cortico-hippocampal network coherence	Functional connectivity analysis	[29]
reduced connectivity between the hippocampus and the striatum	Functional MRI analysis	[31]
functional hypoconnectivity to regions of the default mode network and hyperconnectivity to the lateral occipital cortex	Connectivity and Magnetic Resonance Spectroscopy Study	[32]

## Data Availability

Not applicable.

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
