# Peer review of "Structural and Functional Deviations of the Hippocampus in Schizophrenia and Schizophrenia Animal Models"

_ijms, 2022, doi:10.3390/ijms23105482_

Round 1

Reviewer 1 Report

In this manuscript by Wegrzyn, Juckel and Faissner a very comprehensive review of the structural and functional “deviations” of the hippocampus has been made. The authors very well summarized the data from imaging as well as postmortem studies regarding to macrostructural changes in the hippocampus and then they discussed about the findings in animal models and tried to relate basic research readouts with clinical research readouts.
In summary is a very well written manuscript with several new bibliography combined with classic studies and the resultant message is clear and for sure will be helpful for the readership of IJMS.
I would like to recommend this manuscript for publication but first, I would suggest some minor amendments or changes that I think could help to improve the manuscript.
-First, I think that the initial description of schizophrenia depicted in the manuscript is a bit too classical and should be updated in terms of age of onset and epidemiology since currently the typical onset in the adolescence it is not so adolescence-dependent. As example about what I say can be found here: DOI: 10.1001/jamapsychiatry.2019.3360
-Second, I would suggest adding some papers talking about dendritic spine pathology in the hippocampus in the context of schizophrenia
-Third, I would suggest putting in a table all the human studies in order to make it easier to interpret, including the studies about dendritic spine pathology I just mentioned in the previous point. In this table, I would put the type of alteration (Reduced spines in CA1 pyramidal cells or Reduced volume of the hippocampus, etc… ) plus the reference plus the approach employed (postmortem study or MRI study etc.). This table would support figure 1 or, if the information is enough, it could even substitute it if in the table the authors use the same dimensions or classification: Incomplete inversion patterns, General volume reduction, etc….
-Fourth I found very interesting that the authors found space to talk about interneurons, principal neurons and even microglia. However, I miss some discussion about astroglial cells. Could the authors include a paragraph or some sentences about astroglial alterations in the hippocampus in the context of schizophrenia? Even though there are not too many reports.

Minor points
-In figure 4 I would add a couple of references for each arrow. In other words, for the arrow “optogenetic inhibition” I would put in one side of the sentence a couple of references supporting this statement. This suggestion is just to make the figures a bit more interactive and useful for the reader.
-Maybe is because of my ignorance but the acronym CBV is confusing to me since it seems that refers to the cerebellum but the manuscript is about the hippocampus. 

Reviewer 2 Report

This review of Wegrzyn et al focused on structural and functional alteration of hippocampus in in schizophrenia patients and animal models.

 The manuscript is well written. Specific comments are listed below

  • The authors described only rodent model and no any information about abnormality in the hippocampus of other species (primates, zebrafish)
  • Please specify structural and functional alterations of hippocampus (if present) in rat and mice.  
